# Pyroptosis in Endothelial Cells and Extracellular Vesicle Release in Atherosclerosis via NF-κB-Caspase-4/5-GSDM-D Pathway

**DOI:** 10.3390/ph17121568

**Published:** 2024-11-22

**Authors:** Salman Shamas, Razia Rashid Rahil, Laveena Kaushal, Vinod Kumar Sharma, Nissar Ahmad Wani, Shabir H. Qureshi, Sheikh F. Ahmad, Sabry M. Attia, Mohammad Afzal Zargar, Abid Hamid, Owais Mohmad Bhat

**Affiliations:** 1Department of Biotechnology, School of Life Sciences, Central University of Kashmir, Ganderbal 191201, India; peerzadasalman@cukashmir.ac.in (S.S.); raziarahil119@cukashmir.ac.in (R.R.R.); waninh@yahoo.co.in (N.A.W.); shabirhq@gmail.com (S.H.Q.); zargarma@gmail.com (M.A.Z.); 2Department of Dermatology, Venereology & Leprology, Postgraduate Institute for Medical Education and Research, Chandigarh 160012, India; laveenakaushal1@gmail.com (L.K.); vinsh777@gmail.com (V.K.S.); 3Department of Pharmacology and Toxicology, College of Pharmacy, King Saud University, Riyadh 11451, Saudi Arabia; fashaikh@ksu.edu.sa (S.F.A.); attiasm@ksu.edu.sa (S.M.A.)

**Keywords:** endothelial cells, pyroptosis, atherosclerosis, inflammasome, endothelial dysfunction, inflammation, cytokines

## Abstract

**Background**: Pyroptosis, an inflammatory cell death, is involved in the progression of atherosclerosis. Pyroptosis in endothelial cells (ECs) and its underlying mechanisms in atherosclerosis are poorly understood. Here, we investigated the role of a caspase-4/5-NF-κB pathway in pyroptosis in palmitic acid (PA)-stimulated ECs and EVs as players in pyroptosis. **Methods**: Human umbilical vein endothelial cells (HUVECs) were cultured in an endothelial cell medium, treated with Ox-LDL, PA, caspase-4/5 inhibitor, NF-κB inhibitor, and sEV release inhibitor for 24 h, respectively. The cytotoxicity of PA was determined using an MTT assay, cell migration using a scratch-wound-healing assay, cell morphology using bright field microscopy, and lipid deposition using oil red O staining. The mRNA and protein expression of GSDM-D, CASP4, CASP5, NF-κB, NLRP3, IL-1β, and IL-18 were determined with RT-PCR and Western blot. Immunofluorescence was used to determine NLRP3 and ICAM-1 expressions. Extracellular vesicles (EVs) were isolated using an exosome isolation kit and were characterized by Western blot and scanning electron microscopy. **Results:** PA stimulation significantly changed the morphology of the HUVECs characterized by cell swelling, plasma membrane rupture, and increased LDH release, which are features of pyroptosis. PA significantly increased lipid accumulation and reduced cell migration. PA also triggered inflammation and endothelial dysfunction, as evidenced by NLRP3 activation, upregulation of ICAM-1 (endothelial activation marker), and pyroptotic markers (NLRP3, GSDM-D, IL-1β, IL-18). Inhibition of caspase-4/5 (Ac-FLTD-CMK) and NF-κB (trifluoroacetate salt (TFA)) resulted in a significant reduction in LDH release and expression of caspase-4/5, NF-κB, and gasdermin D (GSDM-D) in PA-treated HUVECs. Furthermore, GW4869, an exosome release inhibitor, markedly reduced LDH release in PA-stimulated HUVECs. EVs derived from PA-treated HUVECs exacerbated pyroptosis, as indicated by significantly increased LDH release and augmented expression of GSDM-D, NF-κB. **Conclusions:** The present study revealed that inflammatory, non-canonical caspase-4/5-NF-κB signaling may be one of the crucial mechanistic pathways associated with pyroptosis in ECs, and pyroptotic EVs facilitated pyroptosis in normal ECs during atherosclerosis.

## 1. Introduction

Pyroptosis is a newly discovered pattern of programmed cellular necrosis often induced by endogenous injury or bacterial and viral infections [1]. Unlike apoptosis, which is a non-inflammatory process that maintains membrane integrity, pyroptosis involves cell swelling and membrane rupture, releasing inflammatory cytokines that alert the immune system. While necroptosis also leads to cell membrane rupture, it is regulated by different signaling pathways, involving specifically RIPK1 and RIPK3 kinases rather than inflammasomes and gasdermin activation, as in pyroptosis. Ferroptosis, another unique type of cell death, is marked by iron-dependent lipid peroxidation rather than the pore-forming proteins seen in pyroptosis [2]. Together, these diverse pathways allow the body to respond to various types of cellular stress and damage. However, pyroptosis plays a critical role during infectious and inflammatory conditions through mobilizing an immune response. Various stimuli, such as nicotine, hyperlipidemia, oxidized modified low-density lipoprotein (Ox-LDL), cholesterol crystals (CCs), and palmitate, promote atherosclerosis and can induce pyroptosis-associated inflammasome and caspases pathway activation through different signaling pathways [3,4,5,6].The molecular mechanisms of pyroptosis mainly include caspase-1-dependent canonical and caspase-4/5/11-dependent non-canonical pathway under different pathological conditions. In the caspase-1-dependent pyroptosis pathway, cells can activate inflammatory vesicles to trigger pyroptosis in response to multiple factors activating the respective inflammasomes (including NLRP3, AIM2, or Pyrin) through the action of pathogen-associated molecular patterns (PAMPs) and damage-associated molecular patterns (DAMPs) [1]. Alternatively, pyroptosis can be induced via the non-canonical pathway by the intracellular detection of lipopolysaccharides (LPS), oxidized phospholipids, and palmitate, which activates caspases-4, 5, and/or 11 (mouse caspase-11) [7,8,9]. Once activated, caspase-4/5/11 cleaves gasdermin D (GSDMD), generating N-GSDMD, which oligomerizes and translocates to the cell membrane to form pores. Although caspase-4/5/11 facilitates the maturation and secretion of IL-1β and IL-18 through the NLRP3/caspase-1 pathway in various cells, the cleavage of GSDMD by caspase-4/5/11 leads to potassium efflux, which induces the assembly of the NLRP3 inflammasome, ultimately resulting in pyroptosis [10]. Literature cites that the endothelium acting as a vascular barrier plays a crucial role in vascular hemostasis and angiogenesis. Under various pathological conditions, such as metabolic disorders, sepsis, and atherosclerosis, pyroptotic endothelium behaves as an inflammatory milieu governing endothelial dysfunction and cell death [11].

Endothelial dysfunction, a basic underlying cause of vascular diseases, initiates atherosclerosis, followed by a cascade of events (accumulation of lipids, fibrous elements, and calcification), including endothelial activation and inflammation. Endothelial activation is characterized by the upregulation of cell adhesion molecules (CAMs), promoting monocyte adhesion and accumulation. Studies showed that oxidized lipids activate NF-κB, a prominent pro-inflammatory transcription factor, that regulates genes involved in early immune, acute-phase, inflammatory responses [12]. NF-κB activation is also shown to activate inflammasomes (including NLRP3), pro-inflammatory cytokines, cell adhesion molecules (CAMs), and matrix metalloproteinases (MMPs), resulting in endothelial dysfunction, atherosclerotic plaque instability, and rupture [13,14].

Several studies demonstrated that low-shear stress result in endothelial cell (EC) pyroptosis, including EC inflammatory phenotype switching via NLRP3/ROS/NF-κB pathways [15,16,17]. Inflammasomes are innate immune system receptors and sensors; NLRP3 recruits Apoptosis-associated Speck-like proteins containing a CARD (ASC) and pro-caspase-1 through various pathways, causing activation of caspase-1, which in turn activates Gasdermin-D (GSDM-D) and regulates the maturation and secretion of pro-inflammatory cytokines such as IL-1β and IL-18 [18]. GSDM-D is a protein which is composed of a 31 kDa N-terminus (GSDMD-N) and a 22 kDa C-terminus (GSDMD-C) connected by a peptide linker. GSDMD-N triggers the rupture of cell membranes and promotes the release of inflammatory factors, cell swelling, and pyroptosis [19]. A recent study in sepsis observed that increased EC expression of caspase-4/5 is required for activation of endothelial pyroptosis, thereby contributing to the development of fulminant pulmonary edema and acute lung injury [20]. Another study showed that human monocytes from obese patients have elevated caspase-4/5 expression and fatty acids that trigger inflammatory response and consequent release of IL-1β/IL-18 [9]. All these studies provided evidence that caspase-4/5 and GSDM-D play an important role in pyroptotic pathway under in vitro and in vivo conditions. Firstly, we explored the role of the NF-κB-caspase-4/5-GSDM-D pathway in endothelial dysfunction and pyroptosis in palmitic acid (PA)-stimulated human umbilical vein endothelial cells (HUVECs) during atherosclerosis.

Studies have shown that pyroptosis of ECs, macrophages, and smooth muscle cells in vascular walls can accelerate atherosclerosis progression [21,22,23]. Recently, it was found that during pyroptosis, the extracellular vesicles (EVs) released contribute to the development of various diseases. Li et al. (2023) revealed that endothelial EVs carrying HIF1A-AS2-induced pyroptosis, vascular inflammation, and accelerated progression of atherosclerosis via ESRRG/NLRP3 [24]. Several studies depicted that EVs, in particular exosomes, are involved in the release of inflammasomes and its products, such as NLRP3 and IL-1β in mammalian cells, which enhanced progression of atherosclerosis [25,26]. Among EVs, small extracellular vesicles (sEVs) or exosomes are widely explored, which are spherical vesicles with a bilayer membrane and an average diameter of 30–150 nm or 40–120 nm [27]. Exosomes transport a variety of molecules called cargo, which mediate intercellular communication by enabling recipient cells to undergo a variety of biological changes [27,28]. The literature cites that during endothelial dysfunction, exosomes increase the expression of adhesion molecules; they also decrease nitric oxide (NO) synthesis in ECs [29,30]. Exosomes released by ECs have been linked to a variety of functions, including atherosclerotic ones that increase thrombosis, inflammation, and arterial stiffness, as well as atheroprotective ones that encourage angiogenesis, EC differentiation, and cell survival [31]. Yuan et al. (2020) revealed that IL-1β-containing EVs isolated from atherogenic-stimulated ECs induced the phenotypic transition of VSMC to synthetic phenotypes, promoting VSMC proliferation and migration and the pathophysiological process associated with neointima formation in atherosclerosis [32]. Therefore, the present study aimed to test the hypotheses of whether inflammatory stimulation by PA led to pyroptosis in ECs via the NF-κB-caspase-4/5-GSDM-D pathway, and whether EVs released by these pyroptotic ECs contribute to pyroptosis in normal HUVECs.

## 2. Results

### 2.1. Effect of Palmitic Acid on Cell Viability, Migration and Lipotoxicity in HUVECs

To determine the cytotoxicity of PA in HUVECs, the HUVECs were treated with different concentrations (25, 50, 100, 200, and 400 µM) of PA for 24 h. The MTT assay was performed to determine cell viability. It was observed that PA markedly deteriorated cell viability in a dose-dependent manner (Figure 1A). Further, we observed PA-induced pyroptosis in HUVECs, which is recently explored cell death in vascular cells during atherosclerosis. Our findings depicted PA significantly increased LDH release, and microscopically showed increased membrane blebbing in a dose-dependent manner in HUVECs as shown in Figure 1B,C. In addition, we observed slow migration of ECs with PA treatment (Figure 2A,B) indicating inverse association with wound healing ability. Moreover, PA significantly increased lipid accumulation in Ox-LDL-stimulated HUVECs as confirmed by oil red O staining (Figure 2C,D). Together, these results indicate that PA is associated with pyroptosis during atherosclerosis.

### 2.2. Palmitic-Acid-Induced Inflammation and Endothelial Dysfunction During Pyroptosis in HUVECs

To confirm PA-induced inflammation, firstly, we determined the levels of inflammatory markers that are associated with endothelial dysfunction. HUVECs were exposed to PA (200 µm) for 24 h, and we observed significantly upregulated mRNA expression of inflammatory caspases such as caspase-4 and caspase-5, inflammatory cytokines such as IL-1β and IL-18 levels, and pyroptotic marker GSDM-D in the PA-treated HUVECs, as shown in Figure 3A–E. Furthermore, we found markedly enhanced NLRP3 and ICAM-1 expressions, as indicated by green and red fluorescence in the PA-treated HUVECs, as shown in Figure 4A,B. Also, significantly increased mRNA expressions of NLRP3 and ICAM-1 was observed in the PA-treated HUVECs as compared to control cells (Figure 4C,D). These findings confirm that inflammation and endothelial dysfunction are associated with pyroptosis in endothelial cells.

### 2.3. Inhibition of Non-Canonical Caspase-4/5 Blocked PA-Induced Pyroptosis in HUVECs

Recently, in sepsis, it was observed that an increased expression of caspase-4/5 is required for endothelial pyroptosis during the development of fulminant pulmonary edema and acute lung injury [20]. In this study, we observed a significantly increased mRNA expression of NF-κB, inflammatory caspases such as caspase-4 and caspase-5, and pyroptotic markers like GSDM-D, IL-1β, and IL-18 levels in PA-treated HUVECs as compared to control (Figure 5A–F). However, pharmacological inhibition of the caspase-4/5 [(caspase-4/5 inhibitor (Ac-FLTD-CMK)] significantly decreased the mRNA expression of the above-mentioned markers in PA-treated HUVECs as compared to PA-only-treated cells. Furthermore, we found that caspase-4/5 inhibition significantly decreased LDH release in PA-induced HUVECs (Figure 6A). Similarly, caspase-4/5 inhibition significantly decreased the protein expression of caspase-4, caspase-5, NF-κB, and GSDM-D in PA-treated ECs (Figure 6B–F). These results provided evidence that the non-canonical caspase (caspase-4/5) pathway contributes to pyroptosis in ECs.

### 2.4. Inhibition of NF-κB Pathway Blocked PA-Induced Pyroptosis in HUVECs

NF-κB is a key mediator of NLRP3-inflammasome-mediated pyroptosis; we tried to explore the relationship between the NF-κB-signaling pathway and caspase-4/5-mediated pyroptosis. We investigated whether the inhibition of NF-κB could attenuate caspase-4/5 expression and EC pyroptosis. ECs with prior treatment with NF-κB inhibitor (TFA) followed by PA stimulation showed significantly decreased mRNA expression of NF-κB, inflammatory caspases such as caspase-4 and caspase-5, and pyroptotic markers like GSDM-D, IL-1β, and IL-18levels as compared to the control (Figure 7A–F). Furthermore, we found that NF-κB inhibition significantly decreased LDH release in PA-induced HUVECs (Figure 8A). In addition, NF-κB inhibition significantly downregulated protein expressions of NF-κB, caspase-4, caspase-5, and GSDM-D in PA-treated ECs (Figure 8B–F). These results indicated that the NF-κB-signaling pathway is associated with non-canonical caspase-4/5-mediated EC pyroptosis.

### 2.5. Contribution of EVs in Endothelial Cell Pyroptosis in PA-Induced HUVECs

EVs are crucial for cell-to-cell communication and vascular remodeling during atherosclerosis. Firstly, we isolated small EVs from ECs and characterized them by SEM and Western blot using exosomal markers (Alix, TSG101), as shown in Figure 9A,B. Secondly, we treated ECs with GW4869, an exosome inhibitor, and inhibiting exosome secretion dramatically reduced LDH release in PA-stimulated HUVECs (Figure 9C). To clarify whether pyroptotic EVs are associated with pyroptosis in normal ECs, we isolated EVs from PA-induced HUVECs and then treated control HUVECs with these pyroptotic EVs (50 µg/mL). We observed pyroptotic EVs significantly augmented LDH release in the control HUVECs, as shown in Figure 9D. Furthermore, significantly increased expression of pyroptotic markers, such as NF-κB and GSDM-D, was found in HUVECs treated with EVs derived from PA-stimulated HUVECs (Figure 9E–G). Moreover, no effect was observed with control EVs isolated from control HUVECs. Together, these results indicate that increased EV release facilitates pyroptosis in normal ECs during inflammation or endothelial injury.

## 3. Discussion

The present study revealed that NF-κB-caspase-4/5-GSDM-D signaling may be one of the crucial mechanistic pathways associated with pyroptosis, and pyroptotic EVs facilitated pyroptosis during atherosclerosis. In HUVECs, under in vitro conditions, firstly, we found that PA stimulation significantly changed the morphology of the HUVECs characterized by cell swelling, plasma membrane rupture, release of pro-inflammatory factors, and increased LDH release, which are features of pyroptosis. PA-treated cells depicted increased accumulation of lipids, as detected by oil red O staining, and inhibited the migration of ECs. Our findings are consistent with earlier studies, which showed that PA, a common saturated fatty acid in circulation, is involved in the development and progression of atherosclerosis and coronary artery disease (CAD) [33]. PA has been shown to activate NLRP3, which in turn enhances ROS generation in macrophages and subsequently weakens the AMPK signaling, which is a negative regulator ROS generation and inflammation [34]. Zeng et al. showed that PA-induced significant pyroptosis, evidenced by significantly elevated mRNA and protein levels of inflammasome markers NLRP3, caspase-1, and IL-1β, as well as cell membrane perforation-driving protein GSDM-D in HepG2 cells [35]. Håversen, L. et al. found that PA increased the production of TNF-α, IL-1β, and IL-8 in human macrophages generated from THP-1 cells [36]. Another study showed that PA stimulates NLRP3 inflammasome activation through the TLR4-NF-κB signal pathway in hepatic stellate cells [37].

Increasing evidence depict vascular endothelial dysfunction, the inner lining of the blood vessels, as the key player in the pathogenesis of atherosclerosis. Free-fatty-acid-mediated endothelial dysfunction involves several mechanisms, including impaired insulin signaling and nitric oxide production, oxidative stress, inflammation, and the activation of the renin–angiotensin system and apoptosis in the ECs. Therefore, targeting the signaling pathways involved in free-fatty-acid-induced endothelial dysfunction could serve as a preventive approach to protect against this pathogenesis and the subsequent complications, such as atherosclerosis. Our study demonstrated that PA-induced endothelial dysfunction significantly increased expressions of NLRP3, ICAM-1, GSDM-D, and pro-inflammatory cytokines IL-1β and IL-18 in the HUVECs during PA-induced pyroptosis. These results are consistent with the different studies which demonstrated that PA is able to induce endothelial dysfunction via regulating NLRP3-mediated pyroptosis [38]. Maloney et al. demonstrated that PA exposure increased superoxide levels and activated TLR4-NF-κB signaling in ECs [39]. Nicolas J. Pillon et al. provided evidence that the saturated fatty acid PA activates the gene expression of IL-1β and activates caspase-4/5 in human monocytes, leading to the release of pro-inflammatory cytokines IL-1β and IL-18 [9].

Previous research on pyroptosis was mostly focused on the function of caspase-1 in the canonical pathway and role of NF-κB in the regulation of canonical pyroptosis via NLRP3 [40,41]. However, the present study in ECs investigated the role of the NF-κB-caspase-4/5-mediated pathway in association with GSDM-D in pyroptosis during atherosclerosis. Our results demonstrate that the expressions of caspase-4, caspase-5, GSDM-D, NF-κB, IL-18, and IL-1β were significantly increased in PA-treated HUVECs both at mRNA and protein levels. Ac-FLTD-CMK, a caspase-4/5 inhibitor, TFA, and the NF-κB inhibitor significantly decreased the PA-induced caspase-4/5, NF-κB, GSDM-D, pro-inflammatory cytokine expression, and LDH release, indicating that the NF-κB-caspase-4/5-GSDM-D signaling pathway is associated with pyroptosis. Consistent with our results, Yang et al. reported that NF-κB regulates Fas-induced cell death via controlling caspase-4 in neuroblastoma cells [42]. Also, Tian et al. showed that caspase-4 expression, regulated by NF-κB, leads to inflammation and apoptosis, resulting in human coronary artery endothelial cells damage [43]. Another study demonstrated that NF-κB regulates non-canonical pyroptosis via influencing the expression of GSDM-D and GSDM-B in THP-1 cells [44]. Additionally, the non-canonical NF-κB pathway triggers NLRP3-inflammasome-mediated endothelial pyroptosis and atherosclerosis via activating caspase-1 and GSDM-D in human aortic endothelial cells [45]. Our results are the first of its kind, which showed that the NF-κB-caspase-4/5-GSDM-D pathway regulates pyroptosis in ECs during atherosclerosis.

In recent years, EVs have been deemed essential mediators of cellular communication, carrying a cargo of various signal molecules involved in inflammatory processes that transfer proteins, lipids, and nucleic acids, therefore altering the target cell’s metabolism in various diseases, including cancer, cardiovascular diseases (CVDs), and neurodegenerative problems. Our findings depicted that blocking of EV/exosome secretion with GW4869, an exosome inhibitor, significantly decreased LDH release in PA-stimulated HUVECs, indicating that increased EV release contributes to pyroptosis in ECs during inflammation or endothelial injury. Consistent with our findings, a study found that treatment with the EV biogenesis/release inhibitor GW4869 prior to heat stress induction reduced hepatocyte cell death, LDH release, and improved liver pathology both in vitro in HepG2 cells and in vivo in C57 male mice [46]. Another study found that blocking exosome secretion with GW4869 reduced pyroptosis and improved uremic cardiomyopathy in wild-type mice [47]. A study showed that blocking exosome release with GW4869 reduces sepsis-induced inflammation and cardiac dysfunction in endotoxemic and cecal ligation/puncture septic mice [48]. Additionally, it has been observed that GW4869 treatment reduces hepatic ischemia–reperfusion-induced pyroptosis and subsequent neuroinflammation in the developing hippocampus region of brain [49].

Finally, our study added to the literature, finding that increased EV secretion during pathological conditions enhanced the disease progression by acting as mediators or delivering cargos of inflammatory milieu and other factors to normal nearby cells. In the present study, we found that EVs derived from PA-treated HUVECs significantly increased the expression of pyroptotic markers, such as GSDM-D, and significantly increased LDH release in the control HUVECs. Several studies have reported that EVs present in the atherosclerotic plaques regulate cellular signaling process associated with atherosclerosis progression, including cell death and inflammation [50,51]. A study by Park, S.J. et al. demonstrated that EVs containing Sphingosine 1-Phosphate Receptors 1 and SPR1/3 have been shown to increase pro-inflammatory cytokine production in macrophages, activating the NF-κB- and p38-MAPK-signaling pathways [52]. Pencheg Li et al. demonstrated that EC-derived EVs harboring HIF1A-AS2 triggered pyroptosis and vascular inflammation in ECs to enhance the development of atherosclerosis by sponging miR-455-5p via ESRRG/NLRP3 both in vitro and in vivo [24]. Another study demonstrated that EVs carrying C-reactive proteins (CRPs) can activate the NF-κB transcription factor, increasing COX1/2 enzyme activity in ECs and leading to inflammation and endothelial dysfunction [53]. A recent study demonstrated that sepsis-exos increased the level of miR-885-5p, decreased HMBOX1, elevated IL-1β and IL-18, and promoted pyroptosis in AC16 cells [54]. In a patient study, serum-derived EVs and macrophages from sepsis patients have elevated levels of HMGB1, which activates the NLRP3 inflammasome and causes liver damage [55].

## 4. Materials and Methods

### 4.1. Reagents and Antibodies

The Ox-LDL (L34357, Thermo Fisher Scientific, Waltham, MA, USA), LDH assay kit (C20300, Thermo Fisher Scientific), palmitic acid (P0500, Sigma-Aldrich, St. Louis, MO, USA), caspase-4/5 inhibitor (7242, Tocris, Bristol, UK), NF-κB inhibitor (17493, Cayman Chemicals, Ann Arbor, MI, USA), exosome release inhibitor (D1692, Sigma-Aldrich, St. Louis, MO, USA), oil red O solution (O1391, Sigma-Aldrich, St. Louis, MO, USA), anti-caspase-4 (4450, Cell Signaling Technology, Danvers, MA, USA), anti-caspase-5 (46680, Cell Signaling Technology, Danvers, MA, USA), anti-Gasdermin-D (69469, Cell Signaling Technology, Danvers, MA, USA), anti-NF-κB (8242, Cell Signaling Technology, Danvers, MA, USA), anti-β-Actin (8457, Cell Signaling Technology, Danvers, MA, USA), anti-Alix (92880, Cell Signaling Technology, Danvers, MA, USA), anti-TSG101 (72312, Cell Signaling Technology), anti-Calnexin (GTX13504, GeneTex, Irvine, CA, USA)

### 4.2. Cell Culture

The HUVECs (C2519A, LONZA, Walkersville, MD, USA) were cultured in the endothelial cell medium (ECM) (CCM029, R&D Systems, Minneapolis, MN, USA) with growth supplements (CCM027, R&D Systems) and 1% penicillin streptomycin in humidified 100% air at 37 °C with 5% CO_2_. Confluent cell cultures of HUVECs were treated with Ox-LDL (50 μg/mL) [56], PA (200 μM/mL), caspase-4/5 inhibitor (Ac-FLTD-CMK) (10 μM/mL) [57], NF-κB inhibitor (Trifluoroacetate salt (TFA)) (20 μM/mL) [58], and sEV release inhibitor (GW4869) (5 μM/mL) [59] for 24 h, respectively.

### 4.3. Preparation of Palmitic Acid

By dissolving PA in ethanol, 100 mM stock solution of fatty acid was prepared. The stock solution was diluted to the working concentrations in a 1% BSA (in complete growth medium) and then incubated at 37 °C for 30 min to allow fatty acid conjugation to BSA. The 1% BSA was prepared by dissolving BSA in the complete growth medium at room temperature, followed by incubation at 37 °C for 20 min. In the present study, PA was used in a pathophysiologically and physiologically relevant concentration range of 25–400 μM.

### 4.4. Cell Viability Assay

The cells were seeded at a density of 3000 cells/mL/well in 96-well microtiter plate and treated with PA in dose-dependent manner at 25, 50, 100, 200, and 400 μM concentrations for 24 h. Then, 2.5 mg/mL of the MTT dye (3-(4,5-dimethylthiazolyl-2)-2,5-diphenyltetrazolium bromide) was added to each well and incubated for 3–4 h. The MTT formazon crystals were dissolved in 100 μL of DMSO, optical density was measured at 540 nm, and the intensity of the dissolved colored formazon product was referred as % age viability and calculated by comparing the absorbance of treated versus untreated cells.

### 4.5. Oil Red O Staining

Prior to the PA (200 µM) treatment, the cells were treated with 50 μg/mL Ox-LDL for 24 h. The media were removed from each well, and the cells were washed gently twice with PBS, followed by fixing with formalin (10%) for 30 min. Cells were washed gently twice with dH_2_O, and isopropanol (60%) was added to each well and incubated for 5 min. Oil red O working solution was added and incubated for 10–20 min and then washed twice with dH_2_O. Finally, cells were stained with hematoxylin, and pictures were taken under a bright field microscope. The oil red O staining was measured semi-quantitatively. After staining with hematoxylin, the cells were washed three times with 60% isopropanol. The oil red O stain was extracted with 100% isopropanol for 5 min with gentle rocking, and absorbance was measured at 492 nm.

### 4.6. Bright Field Microscopy

To examine the morphological change in pyroptotic cells with different concentrations of PA, the cells were seeded in 6-well culture plates at approximately 1 × 10^5^ cells/well, and then after 70–80% confluency, the cells were treated with different concentrations of PA. Static bright field images were captured using EVOS M7000 microscope, Waltham, MA, USA. All image data are representative of at least five randomly selected fields.

### 4.7. Quantitative RT-PCR Analysis

RT-PCR was performed to determine the mRNA expression of *GSDM-D*, *CASP4*, *CASP5*, *NF-κB*, *NLRP3*, *IL-1β*, and *IL-18* using gene specific primers. The primer sequences used were:
**Gene****Primer***GSDM-D*Forward 5′-GTGTGTCAACCTGTCTATCAAGG-3′Reverse 5′-CATGGCATCGTAGAAGTGGAAG-3′*CASP4*Forward 5′-AAGAGAAGCAACGTATGGCAGGAC-3′Reverse 5′-GGACAAAGCTTGAGGGCATCTGTA-3′*CASP5*Forward 5′-GGTGAAAAACATGGGGAACTC-3′Reverse 5′-TGAAGAACAGAAAGCAATGAAGT-3′*NLRP3*Forward 5′-ATTACCCGCCCGAGAAAGG-3′Reverse 5′-TCGCAGCAAAGATCCACACAG-3′*NF-κB*Forward 5′-GCAGCACTACTTGACCACC-3′Reverse 5′-TCTGCTCCTGAGCATTGACGTC-3′*IL-1β*Forward 5′-CCACAGACCTTCCAGGGAATG-3′Reverse 5′-GTGCAGTTCAGTGATCGTACAGG-3′*IL-18*Forward 5′-GATAGCCAGCCTAGAGGTATGG-3′Reverse 5′-CCTTGATGTTATCAGGAGGATTCA-3′*ICAM-1*Forward 5′-AGCGGCTGACGTGTGCAGTAAT-3′Reverse 5′-TCTGAGACCTCTGGCTTCGTCA-3′


Briefly, total mRNA was extracted using TRIzol reagent and then reverse transcribed into cDNA. PCR amplification was performed with SYBR Green PCR Master Mix. GAPDH was used as an internal control. Data were presented as fold change in target genes normalized to the housekeeping gene, GAPDH [13].

### 4.8. Immunofluorescence

To check inflammasome formation, HUVECs were grown on the cover slides. The slides were fixed with 4% paraformaldehyde (PFA) in phosphate-buffered saline (PBS) for 10 min. After this step, cells were rinsed with PBS and incubated with primary anti-NLRP3 and anti-ICAM overnight at 4 °C. Next day, cells were washed with PBS three times and then incubated with Alexa-488 or Alexa-555-labeled secondary antibodies for 1 h at room temperature. Finally, the slides were mounted by DAPI-mounting solution and then analyzed by immunofluorescence. The photos were taken under fluorescent microscope.

### 4.9. Isolation of Extracellular Vesicles

Briefly, HUVECs were cultured, and media were collected and subjected to centrifugation at 300× *g* at 4 °C for 10 min to remove debris and dead cells. Then, the supernatant was filtered through 0.22 μm filters to remove apoptotic bodies. As per manufacturer’s protocol, this filtered supernatant/media and exosome reagent (2:1 ratio), 4478359, Thermo Fisher Scientific, Waltham, MA, USA) was mixed to form a homogeneous solution and incubated overnight at 4 °C. The next day, the suspension was centrifuged at 10,000× *g* for 1 h at 4 °C. The supernatant was discarded, and the EV pellet was resuspended in ice-cold filtered PBS and centrifuged at 10,000× *g* for 10 min (as per manufacturer’s protocol). Finally, the EV samples were ready for use or stored at −80 °C.

### 4.10. Extracellular Vesicle/Exosome Quantification

The EVs were lysed using RIPA buffer (50 mM Tris pH 6.8, 150 mM NaCl, 1 mM EDTA, 1% NP40) by adding 3:1 volume of the solution to the EV sample followed by vigorous mixing. The EV samples were incubated on ice for 30 min before being centrifuged at 13,000× *g* for 5 min at 4 °C. The supernatant was transferred to 1.5 mL Eppendorf tubes, and the protein concentration of the EV samples was measured using a Bradford assay. Following BSA standards, 0.1, 0.25, 0.5, 1, 2, 5, and 10 mg/mL were used as controls. Then, 1 μL of BSA standards were added to designated wells of a 96-well plate in duplicate, and 1 μL of EV protein sample was loaded in triplicate. Following that, 200 μL of the Bradford reagent was pipetted into each well and mixed by pipetting. The plate was incubated at room temperature for two minutes, absorbance was measured at 595 nm, and the relative protein concentration of the EV samples was determined by comparing them to the BSA standards [60].

### 4.11. Scanning Electron Microscopy

The EVs were characterized by scanning electron microscopy (SEM). For this, the EVs were resuspended in filtered PBS and fixed on a thin glass substrate with 3.7% glutaraldehyde. Then, the EV pellet was subjected to dehydration with ascending sequence of ethanol (40%, 60%, 80%, 98%) and incubated at room temperature overnight. The next day, the EV samples were subjected to gold-palladium sputtering and finally analyzed using SEM.

### 4.12. Western Blotting Analysis

Briefly, the cell and EV lysates were prepared and micro-centrifuged. Proteins in the supernatant were extracted; 25 µg protein was loaded per well, fractionated using 12% SDS-PAGE, and transferred onto PVDF membranes. Membranes were blotted with primary antibodies against ALIX, TSG101, Calnexin, GSDM-D, caspase-4, caspase-5, and NF-κB at 4 °C overnight, followed by incubation with anti-rabbit IgG secondary antibodies for 1 h at room temperature. A chemiluminescence technique was used for identification of bands and the band intensity of target proteins. β-actin was used as a loading control [61].

### 4.13. Scratch-Wound-Healing Assay

This assay was used to measure HUVEC migration, as described previously. Briefly, the scratched area was created using a sterile 200 mL pipette tip in 90% confluent cells. Then, the cells were incubated in complete growth medium (10% FBS) in the absence or presence of PA (200 μM/mL) for 24 h. The cells that migrated into the wound surface were determined under microscope at various timepoints. The ratio of cell migration was calculated as the percentage of the remaining cell-free area compared with the area of the initial scratched area [62].

### 4.14. Lactate Dehydrogenase Assay

To quantify cell death, LDH assays were performed according to the manufacturer’s instructions using the LDH assay kit in fresh cell supernatants for greater accuracy. Firstly, a set of triplicate wells were lysed using 10× lysis buffer and incubated for 45 min at 37 °C, 95% humidity, and 5% CO_2_. A reaction mixture was prepared by adding 600 µL of supplied assay buffer to 11.4 mL of supplied substrate stock solution and mixed thoroughly in the dark. Simultaneously, pre-made reaction mixtures were thawed.

In total, 50 µL of supernatants were added to a 96-well flat-bottom plate in triplicate samples. Then, 50 µL of reaction mixture was added to each well, and the plate was incubated for 30 min in the dark at room temperature. Following incubation, 50 µL of supplied stop solution was added to each well. Optical density values were determined by measuring the absorbance at 492 nm using a plate-reading spectrophotometer. Additionally, absorbance was read at 680 nm to account for any background absorbance values. A vehicle control of added stimulants was included in a control set of triplicates to account for spontaneous LDH activity. The percentage of cytotoxicity was calculated as follows for each absorbance:$$\%\;cytotoxicity=\frac{Compound\;treated\;LDH\;activity-Spontaneous\;LDH\;activity}{Maximum\;LDH\;activity-Spontaneous\;LDH\;activity}\times 100\%$$

Then, the values obtained for cytotoxicity determined from the 680 nm wavelength were subtracted from those measured at the 492 nm wavelength to account for background absorbance.

### 4.15. Statistical Analysis

All of the values are expressed as mean ± SD. Data from two groups were analyzed using the two-tailed Student’s *t*-test. Significant differences among multiple groups were examined using one-way ANOVA, followed by a Holm–Sidak test. *p* < 0.05 was considered statistically significant.

## 5. Conclusions

In conclusion, this study elucidates the complex interplay between pyroptosis, inflammation, EC dysfunction, and atherosclerosis, mediated via the NF-κB-caspase-4/5-GSDM-D pathway and pyroptotic EVs. In the present study, the observed morphological cellular changes, such as cell swelling and plasma membrane rupture, along with increased LDH release, further underscore the role of PA in promoting pyroptosis. The inhibition of NF-κB and caspase-4/5 effectively reduced levels of pyroptotic markers and LDH release, suggesting potential therapeutic targets for atherosclerosis. Furthermore, the blockade of EVs/exosome secretion with GW4869 reduced pyroptosis, indicating that EVs are critical mediators in pyroptotic process. Our results highlight the importance of EVs derived from PA-treated HUVECs in facilitating pyroptosis and inflammation in normal ECs, likely through the activation of NF-κB-caspase-4/5-GSDM-D pathway. These findings suggest that targeting the non-canonical caspase pathway and pyroptotic EVs could be a novel strategy to mitigate pyroptotic inflammatory cell death associated with endothelial dysfunction and atherosclerosis and identify the non-canonical caspase pathway and pyroptotic EVs’ potential therapeutic targets for its prevention.

### Future Directions

In future studies, to validate our in vitro findings, suitable animal models will be employed to comprehensively assess the role of pyroptotic EVs and the non-canonical caspase pathway in atherosclerosis. Animal models would allow for exploration of the systemic effects of pyroptotic EVs on vascular remodeling and determine whether blocking EV release or non-canonical caspase activity can effectively prevent endothelial dysfunction and reduce plaque formation or progression in vivo. Additionally, examining how other lipid species influence pyroptotic pathway could broaden our understanding of lipid-driven pyroptosis in vascular pathology, which can lead to other therapeutic targets for the prevention of CVDs, particularly atherosclerosis.

## Figures and Tables

**Figure 1 pharmaceuticals-17-01568-f001:**
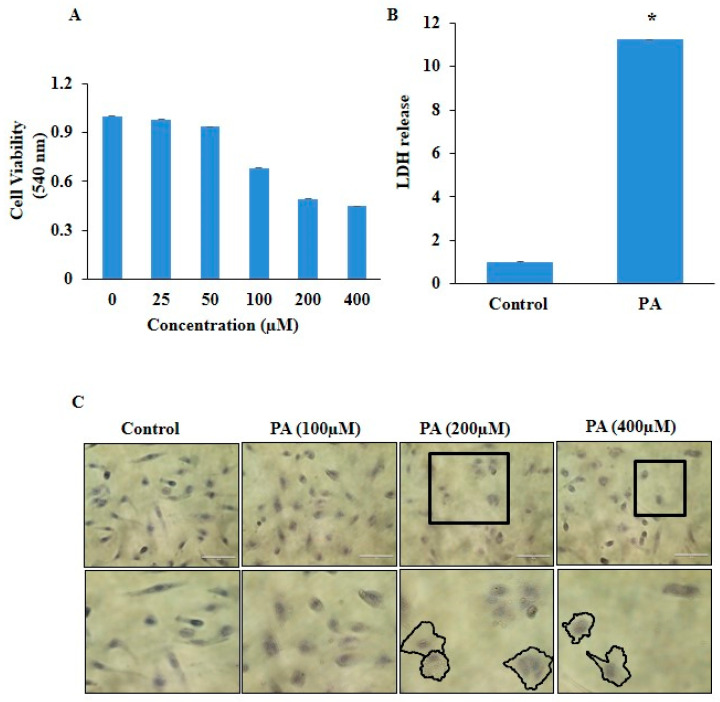
Cell viability and pyroptosis in palmitate-stimulated HUVECs. (**A**) Dose-dependent effect of PA on HUVECs viability compared to untreated control cells by the MTT assay. (**B**) The cellular supernatant LDH level was evaluated with a cytotoxicity detection LDH kit. (**C**) Representative images of HUVECs after their treatment with different concentrations of PA for 24 h. Hematoxylin was used for nuclear staining, and nuclei appear blue. Pyroptotic cells are indicated with square icon and marked structures in second panel indicates blubbed membrane and a swollen structure (Scale bar, 100 μm). *n* = 3. Results are presented as mean ± standard deviation. * *p* < 0.05 vs. the control group.

**Figure 2 pharmaceuticals-17-01568-f002:**
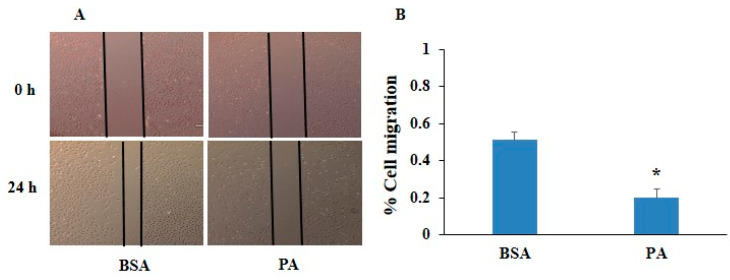
Effect of palmitic acid on cell migration and lipotoxicity in HUVECs. (**A**) The wound-healing assay was performed in the presence of BSA or PA (200 μM) for 24 h (Scale bar: 500 nm) *n* = 3. (**B**) Summarized bar graph showed quantification of cell migration. (**C**) Cells were stained with the oil red O stain, and lipid accumulation was visualized under a microscope after 24 h treatment. Second panel is enlarged portion of square icons in the first panel, and indicates the cells with lipid deposition. Scale bar (100 μm). (**D**) Quantification of the stained lipid droplets was performed using the eluted oil red O stain by measuring absorbance at 495 nm. *n* = 3. Results are presented as mean ± standard deviation. * *p* < 0.05 vs. the control group, $ < 0.05 vs. the PA group, # < 0.05 vs. the Ox-LDL group.

**Figure 3 pharmaceuticals-17-01568-f003:**
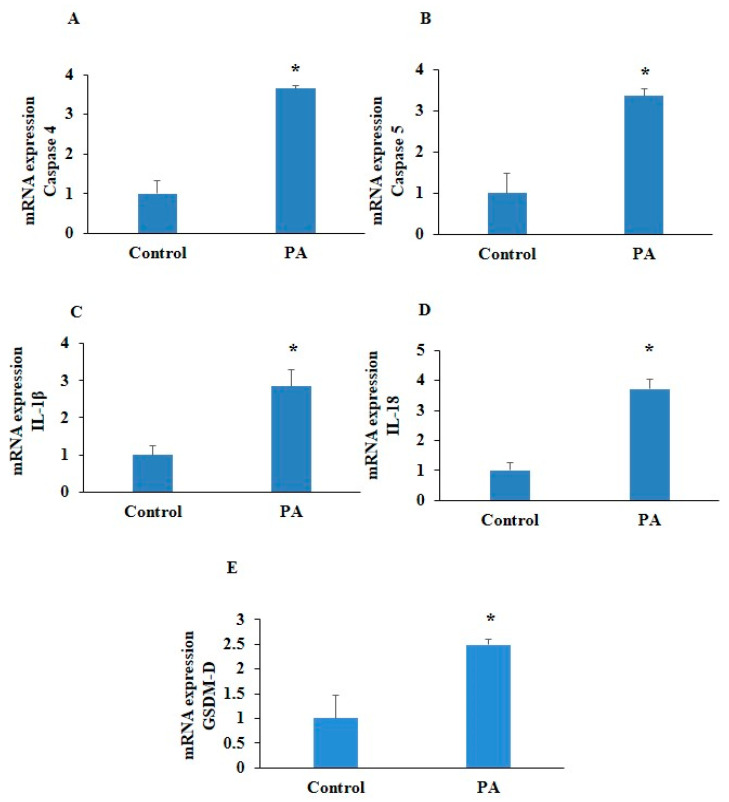
Effect of palmitic acid on inflammatory caspases and pyroptotic markers. Summarized bar graphs showed mRNA expression of (**A**) caspase-4, (**B**) caspase-5, (**C**) IL-1β, (**D**) IL-18, and (**E**) GSDM-D determined by RT-PCR. GAPDH was used as internal control. *n* = 4–5. Results are presented as mean ± standard deviation. * *p* < 0.05 vs. control group.

**Figure 4 pharmaceuticals-17-01568-f004:**
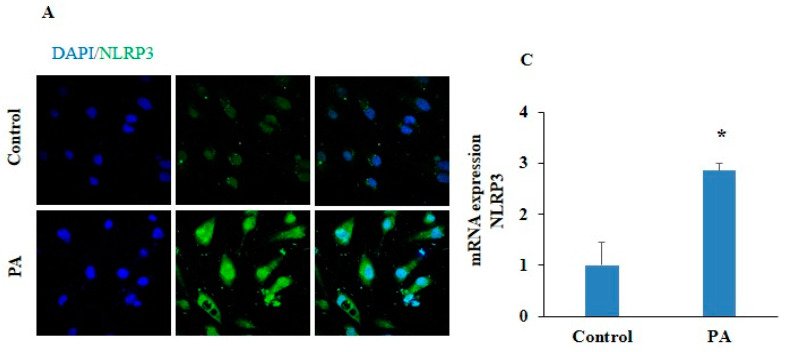
Effect of palmitic acid on endothelial dysfunction. Representative photomicrographs depicted (**A**) NLRP3 (green) and (**B**) ICAM-1 (red) (Scale bar, 100 μm). Summarized bar graph showed mRNA expression of (**C**) NLRP3 and (**D**) ICAM-1 determined by RT-PCR. *n* = 3. GAPDH was used as internal control. Results are presented as mean ± standard deviation. * *p* < 0.05 vs. control group.

**Figure 5 pharmaceuticals-17-01568-f005:**
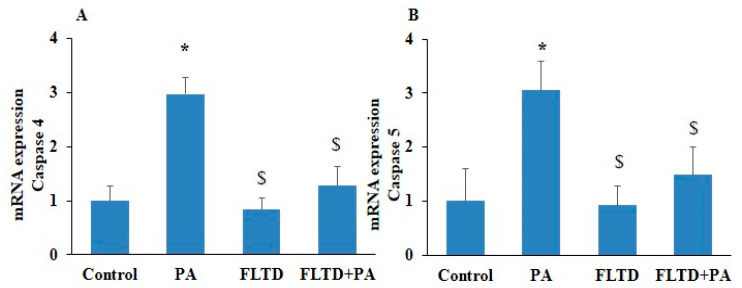
Inhibition of non-canonical caspase-4/5 blocked mRNA expression of pyroptotic pathway markers in PA-induced HUVECs. HUVECs were pre-incubated with caspase-4/5 inhibitor AC-FLTD-CMK (10 µm/mL) for 2 h, then exposed to PA (200 µm) for 24 h. Summarized bar graph showed mRNA expression of (**A**) caspase-4, (**B**) caspase-5, (**C**) NF-κB, (**D**) GSDM-D, (**E**) IL-1β, and (**F**) IL-18, determined via RT-PCR. *n* = 3. GAPDH was used as internal control. *n* = 3–4. Results are presented as mean ± standard deviation. * *p* < 0.05 vs. control group, $ < 0.05 vs. PA group.

**Figure 6 pharmaceuticals-17-01568-f006:**
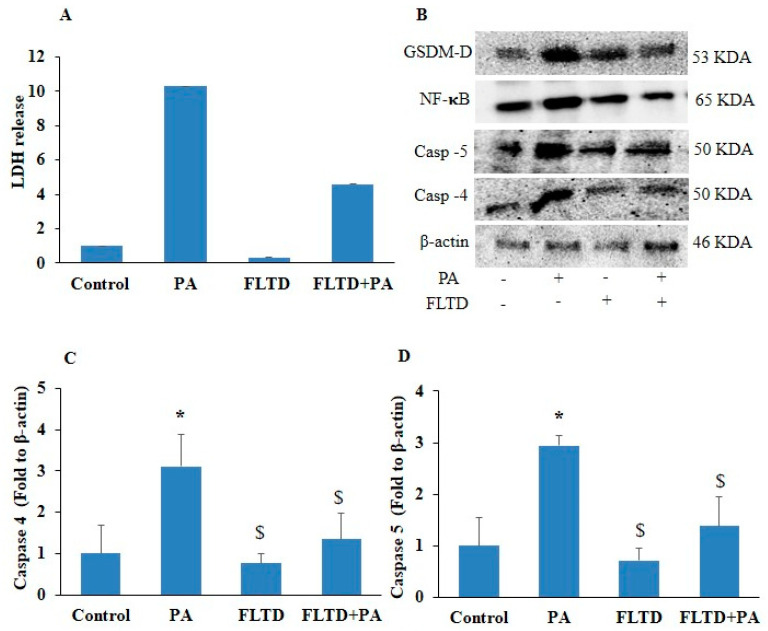
Inhibition of non-canonical caspase-4/5 blocked protein levels of pyroptotic pathway markers in PA-induced HUVECs. HUVECs were pre-incubated with caspase-4/5 inhibitor, AC-FLTD-CMK (10 µm/mL) for 2 h, then exposed to PA (200 µm) for 24 h. (**A**) Lactate dehydrogenase release (LDH). (**B**) Representative Western blot analysis showing the effects of caspase-4/5 inhibition on caspase-4, caspase-5, GSDM-D, and NF-κB protein expression. Summarized data showed the changes in the protein expression of (**C**) caspase-4, (**D**) caspase-5, (**E**) NF-κB, and (**F**) GSDM-D. β-actin was used as an internal control. *n* = 3–4. Results are presented as mean ± standard deviation. * *p* < 0.05 vs. the control group, $ < 0.05 vs. the PA group.

**Figure 7 pharmaceuticals-17-01568-f007:**
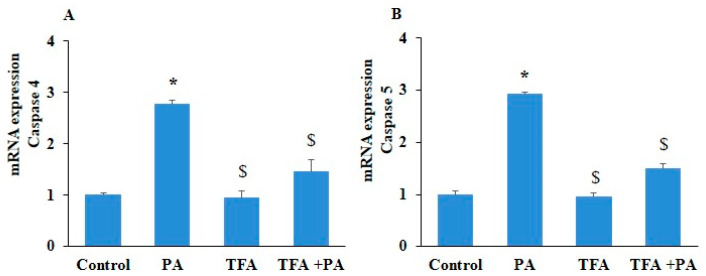
Inhibition of NF-κB blocked mRNA expression of pyroptotic pathway markers in PA-induced HUVECs. HUVECs were pre-incubated with NF-κB inhibitor, trifluoroacetate (TFA) (20 µm/mL) for 2 h, and then exposed to PA (200 µm) for 24 h. Summarized bar graph showed mRNA expression for (**A**) caspase-4, (**B**) caspase-5, (**C**) NF-κB, (**D**) GSDM-D, (**E**) IL-1β, and (**F**) IL-18, determined via RT-PCR. GAPDH was used as internal control. *n* = 3. Results are presented as mean ± standard deviation. * *p* < 0.05 vs. control group, $ < 0.05 vs. PA group.

**Figure 8 pharmaceuticals-17-01568-f008:**
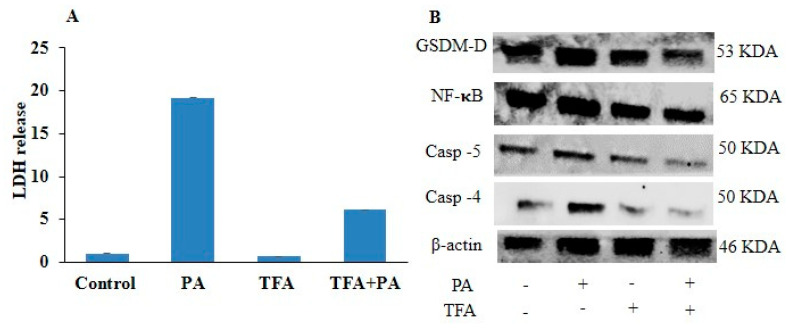
Inhibition of NF-κB blocked protein levels of pyroptotic pathway markers in PA-induced HUVECs. HUVECs were pre-incubated with NF-κB inhibitor, trifluoroacetate (TFA) (20 µm/mL) for 2 h, and then exposed to PA (200 µm) for 24 h. (**A**) Lactate dehydrogenase (LDH) release. (**B**) Representative Western blot analysis showing the effects of NF-κB inhibition on caspase-4, caspase-5, GSDM-D, and NF-κB protein expression. Summarized data show the changes in the protein expression of (**C**) caspase-4, (**D**) caspase-5, (**E**) NF-κB, and (**F**) GSDM-D. β-actin was used as internal control. *n* = 3. Results are presented as mean ± standard deviation. * *p* < 0.05 vs. control group, $ < 0.05 vs. PA group.

**Figure 9 pharmaceuticals-17-01568-f009:**
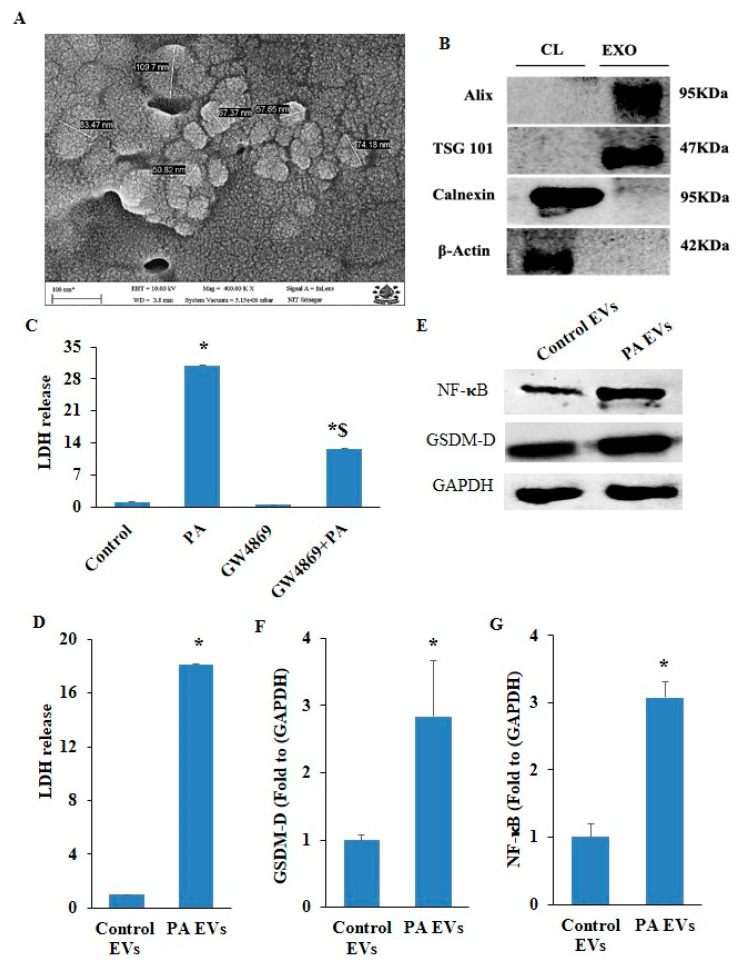
Characterization of small extracellular vesicles/exosomes and their role in pyroptosis in PA-induced HUVECs. (**A**) Representative scanning electron micrographs of sEVs/exosomes isolated from culture medium of HUVECs (Scale bar, 100 nm). (**B**) Representative Western blot analysis of different exosomal protein markers (Alix TSG 101) and cell lysate proteins (Calnexin and β-Actin) collected from HUVECs. (**C**) LDH release was measured in HUVECs, pre-incubated with exosome release inhibitor (GW4869) (5 µm/mL) for 2 h, and then exposed to PA (200 µm) for 24 h. (**D**) HUVECs were co-cultured with EVs isolated from PA-induced HUVECs, and lactate dehydrogenase release (LDH) was measured. (**E**) Representative Western blot analysis showing effects of EVs isolated from PA-induced HUVECs on GSDM-D and NF-κB protein expression. Summarized data showed changes in protein expression of (**F**) GSDM-D and (**G**) NF-κB. *n* = 3. Results are presented as mean ± standard deviation. Data from two groups were analyzed using two-tailed Student’s *t*-test. * *p* < 0.05 vs. control group, $ < 0.05 vs. PA group.

## Data Availability

Data will be available upon request.

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
