# Peer review of "Pyroptosis in Endothelial Cells and Extracellular Vesicle Release in Atherosclerosis via NF-κB-Caspase-4/5-GSDM-D Pathway"

_pharmaceuticals, 2024, doi:10.3390/ph17121568_

Round 1

Reviewer 1 Report

Comments and Suggestions for Authors

The submitted manuscript presents the original research, which aimed to examine the hypothesis that inflammatory activation by palmitic acid induces pyroptosis in endothelial cells through the NF-κB-Caspase 4/5-GSDM-D pathway, and that extracellular vesicles  generated by these pyroptotic cells facilitate pyroptosis in normal human umbilical vein endothelial cells.

This topic is important, and within the scope of the special issue of Pharmaceuticals. Introductions is concise, but at the same time informative. It could be, though, illustrated by adding a suitable figure. Presentation style and scientific novelty are (barely) acceptable.

Taking everything into account I suggest major revision of this work, with the direct questions and points listed below.

Major issues:

First, the similarity level, according to the iThenticate report is very high (55%). Especially the introduction has been mostly directly copied from the other source, shodh.inflibnet.ac.in:8080/jspui/bitstream/20.500.14146/13863/1/salman shamas 21163cukmr002.pdf . The Authors should paraphrase the cited elements.

Second, the number of references (almost 70) must be significantly reduced as it is not a review paper but original article.

A separate “Conclusions” section should be added at the end.

Line 15, “PA” , “HUVEC” abbreviations must be explained. Also, please check similar issues within the whole body of manuscript.

In the introduction the authors should describe the main features of pyroptosis, how does it differ from other types of necrosis?

Figure 1, A and B, the error bars should be added, presenting the SD values.

Figure 9F, how can you explain such large value of SD? It should be discussed.

Minor comments:

Line 33, why “Non-canonical” is with capital “N”?

Line 79, it should be “studies demonstrated”

Line 96, it should be “D pathway”

Line 117 it should be “with neointima”

Line 191, it should be “1β and IL-18 levels”

Also, there’s no need to include the figures after the references section, please remove them.

Author Response

Response to Reviewers

Reviewer #1

Comment 1: The similarity level, according to the iThenticate report is very high (55%). Especially the introduction has been mostly directly copied from the other source, shodh.inflibnet.ac.in:8080/jspui/bitstream/20.500.14146/13863/1/salmanshamas21163cukmr002.pdf. The Authors should paraphrase the cited elements.

Response: Thank you for bringing this matter to our attention. We apologize for any confusion regarding the similarity level. The link provided refers to my own unpublished PhD dissertation, from which we sourced foundational information to build the paper's background. We understand, however, the importance of ensuring originality in our submission and have made necessary modifications to improve paraphrasing and to clarify citations accordingly.

Comment 2: The number of references (almost 70) must be significantly reduced as it is not a review paper but original article.

Response: Thank you for your valuable feedback regarding the number of references. We acknowledge that, as an original article, it would benefit from a more concise reference list. We have carefully reviewed and reduced the number of references to ensure that only the most relevant sources are included.

Comment 3: A separate “Conclusions” section should be added at the end.

Response: Thank you for your suggestion to add a separate "Conclusions" section. We agree that this addition will enhance the clarity of our findings, and we have included a dedicated conclusions section at the end of the manuscript.

Comment 4: Line 15, “PA”, “HUVEC” abbreviations must be explained. Also, please check similar issues within the whole body of manuscript.

Response: Thank you for pointing out the need to clarify abbreviations such as “PA” and “HUVEC” in line 15. We have ensured that these and any other abbreviations are defined upon their first use and have carefully reviewed the entire manuscript to address similar instances.

Comment 5: In the introduction the authors should describe the main features of pyroptosis, how does it differ from other types of necrosis?

Response: Thank you for your suggestion to elaborate on the main features of pyroptosis and its distinctions from other types of necrosis in the introduction. We have incorporated this information to enhance the background and clarity of our study.

Comment 6: Figure 1, A and B, the error bars should be added, presenting the SD values.

Response: Thank you for your observation regarding the error bars in Figure 1, A and B. We would like to clarify that the standard deviation (SD) values are indicated in both figures. However, we ensured they are clearly presented.

Comment 7: Figure 9F, how can you explain such large value of SD? It should be discussed.

Response: Thank you for pointing out this issue. We carefully looked to our data and it seems that out of three experiments (n3), the third experiment has little high values for GSDM-D western blot which led to large SD. However, there is statistically significant difference between the two groups (p<0.05).

Minor Comments:

Comment 1: Line 33, why “Non-canonical” is with capital “N”?

Response: Thank you for pointing out the capitalization issue with "Non-canonical" in line 33. We have corrected it to ensure consistency throughout the manuscript.

Comment 2: Line 79, it should be “studies demonstrated”

Response: Thank you for your observing the error in line 79. Sorry for this typo, we have corrected it to "studies demonstrated."

Comment 3: Line 96, it should be “D pathway”

Response: Thank you for pointing out the correction needed in line 96. We agree that it should be “D pathway,” and we have corrected this typo in the manuscript.

Comment 4: Line 117 it should be “with neointima”

Response: Thank you for pointing out the error in line 117. It was indeed a typo, and we have corrected it to “with neointima.”

Comment 5: Line 191, it should be “1β and IL-18 levels”

Response: Thank you for your suggestion regarding line 191. We agree that it should read “IL-1β and IL-18 levels,” and we have made the correction.

Comment 6: Also, there’s no need to include the figures after the references section, please remove them.

Response: Thank you for your feedback regarding the placement of the figures. We have removed the figures from after the references section as you suggested.

Reviewer 2 Report

Comments and Suggestions for Authors

The authors aim to investigate whether the Caspase 4/5-NFKb pathway is involved in palmatic acid stimulated endothelial cells and whether extracellular vesicles play a role in pryroptosis. The authors concluded that inflammatory non-canonical cascade 4/5-NFKb signaling may be one of the relevant mechanisms associated with pyroptosis in endothelial cells, and that EVs may facilitate pyroptosis in endothelial cells during atherosclerosis.

Overall, the idea is interesting. However there are major concerns:

1) The authors only use HUVEC cells as a model for endothelial cells. It would be important to make sure that their results are the same in other endothelial cell lines and that this is not an event that only happens in HUVEC cells. 

2) All the figures show statistical analysis and error bars, however there is no information in the number of times the experiment was performed. It will necessary to have the number of experiments performed in each figure legend so that we can understand the statistical results and the breath of the results. Also the type of statistical test performed is not mentioned and there is no statistical methods section. Were all the experiments analyzed with the same statistical method? very unlikely, however if this is the case it needs to be described. 

3) The images of the western blots are of very poor quality. The light is saturated in some of images and not in others. It is not explained how they measured intensity of the bands, in order to interpret the protein expression. It is also not explained if they stripped the same membrane or if they did different westerns with the same sample. It is also not explained how much protein was loaded in the western so to load the same amount in each well. 

Overall lots of important details are missing so it makes it hard to know whether the conclusions are sound. 

Comments on the Quality of English Language

Some improvement in writing will be good. 

Author Response

Response to Reviewers

Reviewer #2

Comment 1: The authors only use HUVEC cells as a model for endothelial cells. It would be important to make sure that their results are the same in other endothelial cell lines and that this is not an event that only happens in HUVEC cells.

Response: Thank you for your insightful suggestion regarding the use of HUVEC cells as a model for endothelial cells. We recognize the importance of validating our findings in additional endothelial cell lines to ensure the broader applicability of our results. We have addressed this in our discussion and consider including additional experiments in future studies to confirm that our observations are not exclusive to HUVEC cells. In addition, we currently have another project underway investigating the pyroptotic pathway during vascular calcification using aortic smooth muscle cells. This work will further contribute to understanding the relevance of our findings across different cell types.

Comment 2: All the figures show statistical analysis and error bars, however there is no information in the number of times the experiment was performed. It will necessary to have the number of experiments performed in each figure legend so that we can understand the statistical results and the breath of the results.  Also the type of statistical test performed is not mentioned and there is no statistical methods section. Were all the experiments analyzed with the same statistical method? very unlikely, however if this is the case it needs to be described.

Response: Thank you for your valuable feedback regarding the statistical analysis and the details in the figure legends. We have addressed these points in the Materials and Methods section, where we stated that all values are expressed as mean ± SD. Data from two groups were analyzed using the two-tailed Student’s t-test, while significant differences among multiple groups were examined using one-way ANOVA, followed by the Holm-Sidak test. We have performed almost all experiments minimum three times and maximum five times, indicated by n=3, n=3-4, n=4-5 as mentioned in figure legends.

Comment 3: The images of the western blots are of very poor quality. The light is saturated in some of images and not in others. It is not explained how they measured intensity of the bands, in order to interpret the protein expression. It is also not explained if they stripped the same membrane or if they did different westerns with the same sample. It is also not explained how much protein was loaded in the western so to load the same amount in each well.

Response: Thank you for your observations regarding the western blot images. We apologize for any lack of clarity. To quantify band intensities, we used ImageJ software, ensuring accurate interpretation of protein expression. For each blot, we loaded 25 μg of protein per well to maintain consistency across samples. We have incorporated this change in the material and method section of the manuscript. Additionally, we used the same membrane by stripping and re-probing it for each target protein.

Reviewer 3 Report

Comments and Suggestions for Authors

In the submitted manuscript entitled “Pyroptosis in Endothelial Cells is Associated with Extracellular Vesicle Release during Atherosclerosis: Role of NF-κB-Caspase-4/5-GSDMD pathway” authors Salman et al., exploring the role of inflammatory, non-canonical caspase-4/5- NF-κB signaling pathways associated with pyroptosis in endothelial cells and also described the pyroptotic EVs facilitated pyroptosis in normal ECs during atherosclerosis. The manuscript is well-explanatory and written concisely. I have some minor comments that should be addressed by the authors.

1.     The title of the manuscript should be modified by the authors to make it more concise and simpler.

2.     Authors should make sure that abbreviations is common and must explained before using them in the text.

3.     Metabolomics is crucial for comprehending the biological processes involved. Have the authors checked any metabolomics experiments? Please explain.

4.     It would be more informative if the authors included some in vivo experiments.

5.     How the authors selected the single concentration of PA (200µM) in the most of assay. Did the authors check with another concentration to validate the experiments? Justify.

6.     The manuscript is more than 45 % similar to the published report. It should be less than 10%, or according to journal guidelines.

7. The conclusion part should be concise and add future directions.

8.     The manuscript should be proofread for formatting and grammar issues.

Author Response

Response to Reviewers

Reviewer #3

Comment 1: The title of the manuscript should be modified by the authors to make it more concise and simpler.

Response: Thank you for your suggestion regarding the title. We agree that a more concise and straightforward title would enhance clarity, and we have revised it accordingly.

Comment 2: Authors should make sure that abbreviations is common and must explained before using them in the text.

Response: Thank you for your observation regarding the use of abbreviations. We have reviewed the manuscript carefully and made changes wherever needed to ensure all abbreviations are common and clearly explained upon first use.

Comment 3: Metabolomics is crucial for comprehending the biological processes involved. Have the authors checked any metabolomics experiments? Please explain.

Response: Thank you for your suggestion regarding metabolomics experiments. Although we have not conducted extensive metabolomics analyses in this study, we have measured lactate dehydrogenase (LDH) levels as an indicator related to cell damage and metabolic activity. We appreciate your insight, and we will consider expanding our approach to include additional metabolomics analysis in future studies.

Comment 4: It would be more informative if the authors included some in vivo experiments.

Response: Thank you for your valuable suggestion regarding in vivo experiments. We agree that including in vivo data would enhance the depth of our study. While this work primarily focuses on in vitro findings, we will consider incorporating in vivo experiments in future studies to provide additional insights. Furthermore, we want to mention here that we got recently one project sanctioned (October 2024 ) which will be focused in Rheumatoid Arthritis patients associated with cardiovascular disorder, and pyroptotic signalling and other mechanistic studies will be performed in synovial lining cells (SLCs) known as Synoviocytes and HUVECs.

Comment 5: How the authors selected the single concentration of PA (200µM) in the most of assay. Did the authors check with another concentration to validate the experiments? Justify.

 Response:  Thank you for your question regarding our selection of the PA concentration. To determine the optimal concentration, we conducted a cell viability assay across a range of PA concentrations and identified the IC50 value at 200 µM. Additionally, we validated these findings by performing microscopic analysis at different concentrations to confirm the effects observed based on the cellular morphology.

Comment 6: The manuscript is more than 45 % similar to the published report. It should be less than 10%, or according to journal guidelines.

Response:  We apologize for any confusion regarding the similarity level. The similarity largely arises from content drawn from my own unpublished PhD dissertation, which served as a foundation for building the background of this manuscript. We understand the importance of ensuring originality in our submission and have made the necessary modifications to improve paraphrasing and clarify citations accordingly, in line with the journal’s guidelines.

Comment 7: The conclusion part should be concise and add future directions.

Response:  Thank you for your suggestion regarding the conclusion section. We have revised to make it more concise and have included future directions to highlight potential areas for further research.

Comment 8: The manuscript should be proofread for formatting and grammar issues.

Response: Thank you for pointing out the need for proofreading. We have thoroughly reviewed the manuscript to address any formatting and grammatical issues to ensure clarity and coherence.

Round 2

Reviewer 1 Report

Comments and Suggestions for Authors

The Authors have revised and improved their work. Current version can be accepted for publication.

Reviewer 2 Report

Comments and Suggestions for Authors

The concerns were not addressed properly. 

Comments on the Quality of English Language

No comments. 

Reviewer 3 Report

Comments and Suggestions for Authors

The authors have revised the manuscript as per the reviewer’s suggestions. Although they failed to include the in vivo studies in the present work, it is understandable that it required ethical approval and additional time. I hope they will be included in vivo studies in future experiments and projects.   At present the revised manuscript can be accepted for publication.

Comments on the Quality of English Language

No comments!